# Separate And Diffuse: Using a Pretrained Diffusion Model for Better Source Separation

**Shahar Lutati**[1]    **Eliya Nachmani**[2]    **Lior Wolf**[1]
[1]Department of Computer Science    [2]School of Electrical Engineering
Tel Aviv University
{shahar761,enk100}@gmail.com
wolf@mail.tau.ac.il

## Abstract

The problem of speech separation, also known as the cocktail party problem, refers to the task of isolating a single speech signal from a mixture of speech signals. Previous work on source separation derived an upper bound for the source separation task in the domain of human speech. This bound is derived for deterministic models. Recent advancements in generative models challenge this bound. We show how the upper bound can be generalized to the case of random generative models. Applying a diffusion model Vocoder that was pretrained to model single-speaker voices on the output of a deterministic separation model leads to state-of-the-art separation results. It is shown that this requires one to combine the output of the separation model with that of the diffusion model. In our method, a linear combination is performed, in the frequency domain, using weights that are inferred by a learned model. We show state-of-the-art results on 2, 3, 5, 10, and 20 speakers on multiple benchmarks. In particular, for two speakers, our method is able to surpass what was previously considered the upper performance bound.

## 1 Introduction

Existing speech separation methods face limitations in performance and scalability. We demonstrate that combining pretrained generative and source separation models achieves state-of-the-art results across various sources and datasets, with detailed achievable bound analysis.

Here is a simple algorithm for improving source separation. Given a test mixture $m$ of $C$ speakers, (1) apply a deep neural architecture $B$ to $m$ and obtain multiple approximated sources $\bar{v}_d^i$ for $i = 1...C$, (2) apply a generative diffusion model $GM$ using each of the approximations obtaining $\bar{v}_g^i$, and (3) apply a shallow convolutional neural network $F$ to $\bar{v}_d^i$ and $\bar{v}_g^i$ to obtain mixing weights $[\alpha_i, \beta_i] = F(\bar{v}_d^i, \bar{v}_g^i)$ and combine the two approximations linearly in the frequency domain to obtain the output $\bar{v}^i$.

The need to combine in the frequency domain arises because the reconstructed phase in each segment can be arbitrary and phase compensation is needed. In other words, since $\bar{v}_d$ and $\bar{v}_g$ are inferred by different processes, they may be at different phases.

In our experiments, out of the three networks ($B, GM, F$), we only train network $F$, as this training takes place on the same training set used to train network $B$. The other networks are taken, as is, from published models. Specifically, $B$ is either Gated-LSTM (Nachmani et al., 2020) or SepFormer (Subakan et al., 2021) and $GM$ is the DiffWave network (Kong et al., 2020b), which is trained, in an unsupervised way, on the LibriMix dataset (Panayotov et al., 2015) and WSJ0 (Garofolo, John S. et al., 1993) in order to generate speech signals.

Our empirical results, presented in Sec. 5, demonstrate that across multiple deep architectures and methods, applying a pretrained generative diffusion model, in the most straightforward way, pushes the envelope of results further by a significant gap. Our goal is to shed light on this phenomenon. We show the following two results: (1) the mutual information between the best combination of $\bar{v}_d^i$ and $\bar{v}_g^i$ and the underlying ground truth signal $v^i$ is bounded by twice the mutual information between the mixture and the ground truth signal, (2) we provide a bound for the signal-to-distortion ratio (the

SDR error metric) of such combinations $\bar{\boldsymbol{v}}^i$, which depends on the quality of network $B$ and the mutual information between $m$ and each source $\boldsymbol{v}^i$.

## 2 RELATED WORK

Single-channel speech separation is a fundamental problem in speech and audio processing that has been extensively studied over the years (Logeshwari & Mala, 2012; Martin & Cohen, 2018). Recently, deep learning models have been proposed for speech separation, resulting in a significant improvement in performance compared to traditional methods. Hershey et al. (2016) proposed a clustering method that utilizes trained speech embeddings for separation. Yu et al. (2017) proposed the Permutation Invariant Training (PIT) at the frame level for source separation, while (Kolbæk et al., 2017) extended this approach by proposing the utterance-level Permutation Invariant Training (uPIT). An influential deep learning method for speech separation over the time domain was introduced by Luo & Mesgarani (2018). This method employs three components: an encoder, a separator, and a decoder. Subsequently, in Conv-Tasnet the separator network was replaced with a fully convolutional model, using a block of time-depth separable dilated convolution (Luo & Mesgarani, 2019). Conv-Tasnet was scaled by training several separator networks in parallel to perform an ensemble (Zhang et al., 2020). Dual Path RNN blocks were used to reorder the encoded representation and process it across different dimensions (Luo et al., 2019). The so-called MulCat blocks were presented by Nachmani et al. (2020) as a way to eliminate the need for a masking sub-network.

One of the limitations of current methods is their inability to effectively train neural networks for a large number of speakers, due to the reliance on Permutation Invariant Training (PIT) methods, which have a time complexity of $O(C!)$, where $C$ is the number of speakers. Instead, one can use a permutation-invariant training method that employs the Hungarian algorithm, reducing the time complexity to $O(C^3)$ (Dovrat et al., 2021).

Lutati et al. (2022) introduced an upper bound for audio source separation. By dividing the speech signal into short segments of sounds, a known distribution is used to describe the signal. Then, using the relation between mutual information and Cramer Rao lower bound, the authors managed to demonstrate an upper bound for speech separation for any deterministic model. This bound depends on the mutual information between the mixture and the sources. As the number of sources increases, the mutual information decreases, and so does the upper bound. The recent advent of very successful randomized generative models points in a new research direction: while the previous bound is applicable to any deterministic processing, it remains an open question whether it holds for generative models too.

The Diffusion Probabilistic Model has been successfully applied to various domains, such as time series and images (Sohl-Dickstein et al., 2015). A major limitation of this model is that it requires a significant number of iterative steps to generate valid data samples. This was addressed by a diffusion generative model based on Langevin dynamics and the score matching method (Song & Ermon, 2019). This model estimates the Stein score function (Liu et al., 2016), which is the gradient of the logarithm of data density, and uses it to generate data points. Generative neural diffusion processes for speech generation were developed based on score matching (Chen et al., 2020; Kong et al., 2020b). These models have achieved state-of-the-art results for speech generation, demonstrating superior performance compared to well-established methods, such as Wavernn (Kalchbrenner et al., 2018), Wavenet (Oord et al., 2016), and GAN-TTS (Bińkowski et al., 2019).

Jayaram & Thickstun (2020) introduced the use of generative models as priors for the separation of different sources from a mixture. This was tested on mixtures of images such as MNIST (LeCun & Cortes, 2010) and LSUN (Yu et al., 2015) datasets. DiffSep (Scheibler et al., 2022) performs audio separating using a diffusion model. The underlying SDE is an affine transformation of a mixing matrix $P$. While the authors showed the ability to separate sources from a single channel, their result falls behind deterministic models by a large margin. Although they set the ground for source separation using generative models, neither work presented state-of-the-art performance, nor proposed a bound for generative methods.

We are unaware of similar generalization bounds developed for diffusion models. Our method can be applied to other domains, such as image denoising and conditional generation, given suitable data on the underlying distribution of the training data. For example, for image denoising, there are

known results from the field of natural image statistics (Weiss & Freeman, 2007) that state that while segmenting the image into patches a Gaussian Scale Mixture is able to model the statistics of the underlying natural images. A similar derivation to what is done in our work for audio would obtain an upper bound for the maximal improvement achievable given the noise prior.

## 3 ANALYSIS

Let $m$ be a mixture of $C$ speakers, each with a ground truth signal $v^i$, $i = 1 \ldots C$. Let $B$ be a source separation model that returns $C$ signals and $GM$ be a vocoder diffusion model that is trained on the domain of clear, single-speaker voice signals. The method described at the beginning of Sec. 1 can be summarized by the following set of equations, where STFT and iSTFT are the Short-Time Fourier transform and its inverse, respectively.

$$[\bar{\boldsymbol{v}}_d^1, \bar{\boldsymbol{v}}_d^2, \ldots, \bar{\boldsymbol{v}}_d^c] = B(m) \tag{1}$$

$$\forall i \in [c] \begin{cases} \bar{\boldsymbol{v}}_g^i = GM(\bar{\boldsymbol{v}}_d^i) & (2) \\ \bar{V}_g^i, \bar{V}_d^i = \text{STFT}(\bar{v}_g^i), \text{STFT}(\bar{v}_d^i) & (3) \\ [\alpha_i, \beta_i] = F(\bar{\boldsymbol{v}}_d^i, \bar{\boldsymbol{v}}_g^i) & (4) \\ \bar{\boldsymbol{v}}^i = \text{iSTFT}(\alpha_i \odot \bar{\boldsymbol{v}}_d^i + \beta_i \odot \bar{\boldsymbol{v}}_g^i) & (5) \end{cases}$$

Eq. 1 applies $B$ to separate the mixed signal. Eq. 2 applies the vocoder to denoise each output of $B$ separately. Eq. 4 computes the combination weights of the output of $B$ and the corresponding output of $GM$. Finally, the combination is performed by linearly mixing in the spectral domain in Eq. 5. Where $\odot$ is the Hadamard product.

As we show in Sec. 5, this simple method empirically surpasses the current state-of-the-art. Evidently, when applied correctly, a vocoder model can suppress the errors that the deterministic source separation model has. Below, we analyze the source of this improvement and derive an upper bound for the level of improvement.

**Definition 3.1** (Signal-to-Distortion Ratio (SDR)). Let the error $\epsilon$ be defined as the difference between the source and the estimated source. The Signal-to-Distortion Ratio (SDR) is $SDR = 10log_{10} \frac{Var(v)}{Var(\epsilon)}$.

Given a mixture of sources, $m$, the ground truth signal $v$, and the estimated signal, $\bar{v}_d$, (Lutati et al., 2022) have found that

$$SDR(\boldsymbol{v}, \bar{\boldsymbol{v}_d}) \leq 10log_{10} \left( \frac{L}{w} \cdot Var(\boldsymbol{v}) \cdot I(\boldsymbol{m}_r; \boldsymbol{v}_r) \right) \tag{6}$$

where $\bar{\boldsymbol{v}_d}$ is the estimation of the deterministic backbone network $B$, $L$ is the length of the signal, $w$ is the segment width, $v$ is the ground truth source, $m$ is the mixture and the subscript $r$ describes the mixture and the ground truth signal in the r-th segment.

Furthermore, (Lutati et al., 2022) show that the mutual information between the deterministic estimation and the source is upper bounded by the mutual information between the source and the mixture.

$$I(\boldsymbol{v}_r; \bar{\boldsymbol{v}}_{dr}) \leq I(\boldsymbol{v}_r; \boldsymbol{m}_r) \tag{7}$$

In what follows, we first lay down the basis for upper bounding the combination of the generative and the deterministic signal. Next, by modeling the noise that is added to the generated signal, the mutual information between the combination of the signals and the sources is bounded. Finally, by applying the changes to Eq. 6, a new upper bound is found.

**Lemma 3.2** (Mutual Information Chain Rule). *For any random variables $X,Y,Z$ the following chain rule holds $I(X;Y,Z) = I(X;Z) + I(X;Y|Z)$.*

See proof in the supplementary. Using Lemma. 3.2 for the source, $\boldsymbol{v}_r$, the deterministic estimation, $\bar{\boldsymbol{v}}_{dr}$, and the generative estimation, $\bar{\boldsymbol{v}}_{gr}$, we have,

$$I(\boldsymbol{v}_r; \bar{\boldsymbol{v}}_{dr}, \bar{\boldsymbol{v}}_{gr}) = I(\boldsymbol{v}_r; \bar{\boldsymbol{v}}_{dr}) + I(\boldsymbol{v}_r; \bar{\boldsymbol{v}}_{gr}|\bar{\boldsymbol{v}}_{dr}) \tag{8}$$

Plug Eq. 8 to Eq. 7, we have the following bound

$$I(\boldsymbol{v}_r; \bar{\boldsymbol{v}}_{dr}, \bar{\boldsymbol{v}}_{gr}) \leq I(\boldsymbol{v}_r; \boldsymbol{m}_r) + I(\boldsymbol{v}_r; \bar{\boldsymbol{v}}_{gr}|\bar{\boldsymbol{v}}_{dr}) \tag{9}$$

Our goal is now to bound the maximal mutual information achievable between the source, $\boldsymbol{v}_r$, and the generative signal, $\bar{\boldsymbol{v}}_{gr}$.

### 3.1 GENERATIVE BOUND

The inherent noise of GM is modeled as additive noise. For any practical GM there is an inherent error in reconstructing a perfect signal from a perfect prior.

The pretrained generative model (GM) is optimized to generate samples $\bar{\boldsymbol{v}}_{gr}$ that minimize the ELBO loss when given a prior for $\boldsymbol{v}_r$. Denote the observed prior as $p(\boldsymbol{v}_{dr})$. GM generates $\bar{\boldsymbol{v}}_{gr}$ that it is the optimizer for ELBO loss: $p(\bar{\boldsymbol{v}}_{gr}) = argmin_{p(\bar{\boldsymbol{v}}_{gr})}ELBO(p(\bar{\boldsymbol{v}}_{dr}), p(\bar{\boldsymbol{v}}_{gr}))$.

Given the aforementioned upper-bound $I(\boldsymbol{v}_r; \bar{\boldsymbol{v}}_{dr})$, Eq. 7, it follows that $p(\bar{\boldsymbol{v}}_{gr}) \approx p(\bar{\boldsymbol{v}}_{dr})$ when $I(\boldsymbol{v}_r; \bar{\boldsymbol{v}}_{dr}) \leq I(\boldsymbol{v}_r; m)$. This implies that the $\bar{\boldsymbol{v}}_{gr}$, will sample from the same distribution as $\bar{\boldsymbol{v}}_{dr}$. Using Eq. 9 We have, $I(\boldsymbol{v}_r; \bar{\boldsymbol{v}}_{gr})$ is upper bounded by $I(\boldsymbol{v}_r; \boldsymbol{m}_r)$, and $I(\boldsymbol{v}_r; \bar{\boldsymbol{v}}_{gr}, \bar{\boldsymbol{v}}_{dr})$.

$$I(\boldsymbol{v}_r; \bar{\boldsymbol{v}}_{gr}) \leq I(\boldsymbol{v}_r; \boldsymbol{m}_r) + I(\boldsymbol{v}_r; \bar{\boldsymbol{v}}_{gr}, \bar{\boldsymbol{v}}_{dr}) \leq 2 \cdot I(\boldsymbol{v}_r; \boldsymbol{m}_r) \tag{10}$$

Let us assume the added noise is Gaussian noise. From (Cover & Thomas, 2006), we have that among all distributions Gaussian noise is the worst-case additive noise in terms of mutual information, that is, normal additive noise will degrade most of the information in the generated signal. As shown below, for the pretrained GM, the worst case coalesces with the best case, where the inequality in Eq. 10 becomes equality. Thus, making Eq. 10 tight.

For additive white Gaussian noise (AWGN) it holds that $\bar{\boldsymbol{v}}_{gr} = x + n, n \sim \mathcal{N}$ where $x$ is the desired part of the signal, based on the prior, and $n$ is the normal noise. $Pr(n) = exp(-\frac{n^2}{2\sigma^2})/\sqrt{2\pi\sigma^2}$, where $\sigma^2$ is the variance of the additive noise. Lutati et al. (2022) show empirically that for short segments of 20[ms] the audio is distributed in the time domain as a Laplace distribution, $x \sim \mathcal{L}aplace$. Therefore, $Pr(x) = exp(-|\frac{x}{\sqrt{2}}|)/\sqrt{2}$.

Computing explicitly the mutual information we obtain

$$I(x; \bar{\boldsymbol{v}}_g) = \int_x \int_{\bar{\boldsymbol{v}}_g} Pr(x, \bar{\boldsymbol{v}}_g) \log\left(\frac{Pr(x, \bar{\boldsymbol{v}}_g)}{Pr(x)\,Pr(\bar{\boldsymbol{v}}_g)}\right) dx\, d\bar{\boldsymbol{v}}_g\,, \tag{11}$$

where $Pr(x)$ is the probability of $x$, $Pr(\bar{v}_g)$ is the probability of the generated voice, and $Pr(x, \bar{\boldsymbol{v}}_g)$ is the joint probability of the signals. The joint probability can be written explicitly as $Pr(x, \bar{\boldsymbol{v}}_g) = Pr(\bar{\boldsymbol{v}}_g|x)Pr(x)$. Plugging in the known distributions reads

$$Pr(x, \bar{\boldsymbol{v}}_g) = \frac{exp(-\frac{1}{2}(\frac{\bar{\boldsymbol{v}}_g - x}{\sigma})^2)}{\sqrt{2\pi\sigma^2}} \cdot \frac{exp(-|\frac{x}{\sqrt{2}}|)}{\sqrt{2}} \tag{12}$$

The distribution of $\bar{v}_g$ can be formulated using marginalization of the joint probability, $Pr(x, \bar{\boldsymbol{v}}_g)$

$$Pr(\bar{v}_g) = \int_x Pr(x, \bar{v}_g)\, dx \tag{13}$$

Recall that the generated voice without noise can at best (maximum) obtain the mutual information between the mixture and the sources. Let $\rho$ be the ratio between the mutual information of the source and the generative estimation, and the mutual information of the source and mixture, i.e., $\rho = \frac{I(\boldsymbol{v}_r, \bar{v}_g)}{I(\boldsymbol{v}_r, \boldsymbol{m}_r)}$. The factor $\rho$ is computed numerically from equation Eq. 11, by integrating the double integral. Figure 1 depicts the ratio $\rho$ as defined above. This ratio is computed for

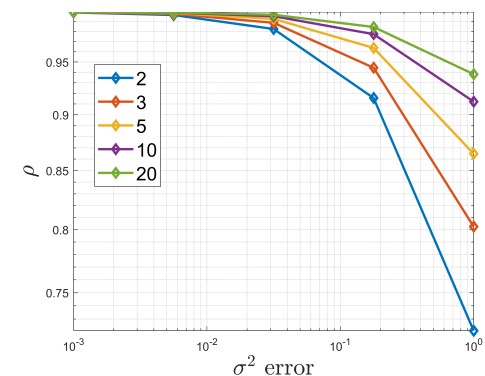

Figure 1: $\rho$ as a function of the noise variance.

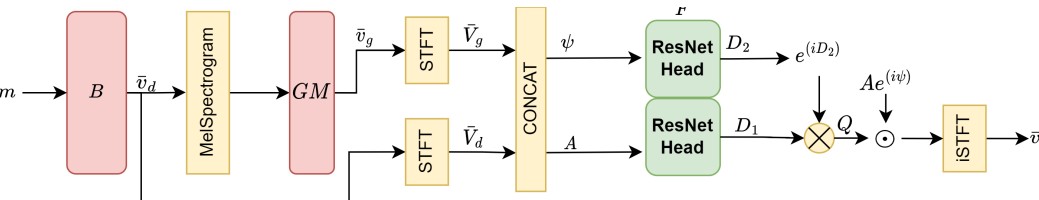

Figure 2: The suggested system architecture. Red blocks indicate pretrained models, Yellow blocks indicate non-learned operations, and Green blocks indicate learned neural models.

numerous sources. Each trial is with a different color. Note that for low-quality generative models (i.e, $\sigma^2 \geq 10^{-2}$) the ratio degrades at a faster rate when the number of sources is lower. Recall that when the number of sources is larger, the mixture tends to have a normal distribution. Thus, the addition of another independent noise with a smaller variance has a negligible effect in terms of the mutual information (which is already low). Observe that $\rho$ is approximately one when $\sigma^2 \leq 10^{-2}$. Since, for any well-functioning pretrained GM, the error in reconstructing the audio signal is below $10^{-2}$ (Kong et al., 2020b; Chen et al., 2020; 2021), we obtain that

$$\rho(\sigma^2 \leq 10^{-2}) \approx 1 \quad \forall C \in [2, 20] \tag{14}$$

Therefore, rewriting Eq. 10 with the definition of $\rho$ reads,

$$\rho I(\boldsymbol{v}_r; \boldsymbol{m}_r) = I(\boldsymbol{v}_r; \bar{\boldsymbol{v}}_g) \leq I(\boldsymbol{v}_r; \bar{\boldsymbol{v}}_{dr}) \leq I(\boldsymbol{v}_r; \boldsymbol{m}_r). \tag{15}$$

Using the result of Eq. 14 one obtains,

$$I(\boldsymbol{v}_r; \boldsymbol{m}_r) \approx I(\boldsymbol{v}_r; \bar{\boldsymbol{v}}_g) \leq I(\boldsymbol{v}_r; \bar{\boldsymbol{v}}_{dr}) \leq I(\boldsymbol{v}_r; \boldsymbol{m}_r), \tag{16}$$

which implies that the bound is tight. Also, from the bound, under the reasonable assumptions made, the deterministic approximation and the generative approximations would have similar error magnitudes.

Finally, updating Eq. 6 with the new bound obtains,

$$SDR(v, \bar{\boldsymbol{v}}) \leq 10 log_{10}\left(\frac{L}{w} \cdot Var(\boldsymbol{v}) \cdot I(\boldsymbol{m}_r, \boldsymbol{v}_r)\right) + 3.0 \tag{17}$$

One can see a maximum addition of 3dB to the upper bound when combining both generative results with deterministic processing. Note that this bound is tight in the sense that the worst-case additive independent noise is taken into account and still the achievable mutual information is at maximum.

The upper bound assumes sequential processing of uncorrelated chunks and does not hold for mixed chunks processing. An approach such as TF-GridNet Wang et al. (2023) processes the signal entirely but has limitations such as being unable to deal with non-stationary segments, such as silence. Also, such methods need to be retrained for every signal length and are only able to handle relatively short signals.

## 4 METHOD

Given a mixture signal $m$, the deterministic estimation $\bar{v}_d$ is obtained using the backbone network $B$. Then, a Mel-Spectrogram is computed over all $\bar{v}_d$ estimations.

$$Mel(\bar{\boldsymbol{v}}_d) = MelSpectrogram(\bar{\boldsymbol{v}}_d) \tag{18}$$

Using a pretrained vocoder, $GM$, specifically DiffWave, and the priors obtained from the deterministic backbone $B$, a generative estimation is obtained $\bar{v}_g$.

$$\bar{\boldsymbol{v}}_g = GM(Mel(\bar{\boldsymbol{v}}_d)) \tag{19}$$

Since the generative vocoder is not given any phase information, the generated signal has a phase shift that can vary over time. To combine both estimations, an alignment procedure is employed.

In order to align the signals, both estimations are transformed into the frequency domain, where the aligning operation dual is multiplication by a phasor. The transformation is done through a short-time Fourier transform with $N_{\text{FFT}}$ frequency bins and $K$ segments.

Denote the spectrogram of $\bar{v}_d^i$ and $\bar{v}_g^i$, as $\bar{V}_d^i, \bar{V}_g^i$ respectively, $\bar{V}_d^i, \bar{V}_g^i \in \mathbb{C}^{N_{\text{FFT}} \times K}$. The absolute phase of $\bar{V}_g$ is of no importance; what is important is its trend over short segments. The phase of $\bar{V}_d$ and the relative phase between $\bar{V}_d$ and $\bar{V}_g$ is concatenated into a 2-channel tensor, $\psi \in \mathbb{R}^{C \times 2 \times N_{\text{FFT}} \times K}$, i.e.,

$$\psi = Concat(\angle \bar{V}_d, \angle(\bar{V}_g \odot \bar{V}_d^*)), \tag{20}$$

where $\odot$ is the Hadamard product, the star superscript denotes conjugate, and the following definition is used.

**Definition 4.1.** The angle, denoted as $\angle$, of a complex number is computed by

$$\angle X = tan^{-1}(\frac{Im(X)}{Re(X)}) \tag{21}$$

where $X \in \mathbb{C}$, $Im$ is the imaginary part, and $Re$ is the real part.

In addition to the relative phase tensor $\psi$, the magnitude of both $\bar{V}_d^i$ and $\bar{V}_g^i$ is concatenated into tensor, $A$,

$$A = Concat(|\bar{V}_d|, |\bar{V}_g|) \tag{22}$$

The alignment network that we train $F$ is a dual 6-layer convolutional neural network, with residual connections (He et al., 2016), termed ResNet Head in Fig. 2. The magnitude and phase heads share the same hyperparameters. The processed tensors are combined into the complex factor $Q \in \mathbb{C}^{C \times 2 \times N_{\text{FFT}} \times K}$,

$$D_1, D_2 = F(A, \psi) \tag{23}$$

where $D_1$ and $D_2$, both in $\mathbb{R}^{C \times 2 \times N_{FFT} \times K}$, represent the magnitude and phase, respectively, of a complex tensor $Q$. This dual representation is used as an effective representation for complex tensors. $Q$ is then computed explicitly by combining the magnitude and phase

$$Q = D_1 \cdot exp(-jD_2) \tag{24}$$

Below, we denote the different channels of $Q$ by a subscript $i$. Define as $\alpha = Q_1 \in \mathbb{C}^{C \times 1 \times N_{FFT} \times K}$ and the second channel as $\beta = Q_2$.

$\alpha$ and $\beta$ are the coefficients used in Eq. 5. Written more explicitly, the weighted sum of $\bar{V}_d$ and $\bar{V}_g$ is computed as

$$\bar{V} = \alpha \odot \bar{V}_d + \beta \odot \bar{V}_g \tag{25}$$

Finally, the time-domain signal is obtained by employing the inverse short-time Fourier transform.

$$\bar{v} = iSTFT(\bar{V}) \tag{26}$$

**Objective Function**    Following (Luo & Mesgarani, 2019; Subakan et al., 2021; Lutati et al., 2022; Scheibler et al., 2022), the objective function is the Scale-Invariant SDR. The scale-invariant SDR is agnostic to the scale of the estimated signal, and also to spurious errors, as presented in (Le Roux et al., 2019). First, project the source onto the estimated signal.

$$\tilde{v} = \frac{<v, \bar{\boldsymbol{v}}> v}{||v||^2} \tag{27}$$

Second, compute the normalized error

$$\tilde{e} = \bar{\boldsymbol{v}} - \tilde{v} \tag{28}$$

Third, the scale-invariant SDR is computed as follows,

$$SI - SDR(v, \bar{\boldsymbol{v}}) = 10log10\left(\frac{||\tilde{v}||^2}{||\tilde{e}||^2}\right) \tag{29}$$

For a large number of speakers ($C \geq 10$), the Hungarian algorithm obtains better results than the permutation invariant loss, by explicitly assigning the different estimated sources to the ground truth sources (Dovrat et al., 2021). Therefore, network $F$ is trained using the assignment obtained by the Hungarian method for the deterministic voice separation network.

## 5 EXPERIMENTS

For all datasets, the same alignment network architecture is employed, with the same hyper-parameters. It is a CNN with six layers, $3 \times 3$ kernels, and 32,32,64,64,64 hidden channels. For Librimix and WSJ0 datasets the DiffWave was trained separately over the training sets' sources (no mixing). The pretrained models for separation are taken from the official publication when available and from HuggingFace hub otherwise. The optimization procedure is done with Adam (Kingma & Ba, 2015) optimizer with a learning rate of 1E-3 and a batch size of 3. The setting involved 3 A5000 GPUs.

**Datasets**     The LibriSpeech dataset (Panayotov et al., 2015) is a large corpus of read English speech, designed for training and evaluating automatic speech recognition systems. It consists of over 1000 hours of audio from audiobooks read by professional and non-professional speakers. The Wall Street Journal (WSJ0) (Garofolo, John S. et al., 1993) dataset is a collection of read English speech samples and is widely used by the research community. The dataset includes a total of 80 hours of audio.

Following previous work (Subakan et al., 2021; Dovrat et al., 2021), mixtures are generated by a random process. The speakers are divided into training speakers and test speakers. A training- or a test sample is created by combining random speakers out of the respective set of speakers, with random SNR values between $0 - 5$ dB.

**Baseline methods**     We compare our method to the state-of-the-art methods in voice separation, including DiffSep (Scheibler et al., 2022) that employs diffusion models, SepFormer (Subakan et al., 2021), which is a transformer model, and SepIt (Lutati et al., 2022), which extends the Gated-LSTM method (Nachmani et al., 2020) by running it in an iterative manner.

As a generative model our experiments employ a pretrained DiffWave model (Kong et al., 2020b).

In addition to running our method, we perform an ablation that is aimed at verifying the need for a learned combination model $F$. In it, we align (recover the phase) the generative approximation with the deterministic one and then average the two. Another ablation is using the GAN-based model HiFiGAN (Kong et al., 2020a), trained on WSJ0, which is a deterministic generative model (no noise is given to the generator). Since this generative model is deterministic the previous upper bound applies and we verify experimentally that no significant improvement is obtained.

First, the cross-correlation function is used as a measure of similarity. In the frequency domain, the cross-correlation dual is computed using conjugate multiplication.

$$\bar{R} = \bar{V}_d \odot \bar{V}_g^* , \tag{30}$$

where $\bar{R}$ is the frequency domain. For each segment, the inverse Fourier transform is conducted, and then the time for which the signal is at maximum is searched.

$$t_i^* = \arg\max(IFFT(\bar{R}[I]) \quad \forall i \in \{1, \ldots, K\} , \tag{31}$$

where $t_i^*$ is the time delay within the i-th segment, $\bar{R}[I]$ is the i-th frequency domain cross correlation segment.

A complex phase shift corresponding to the time delay found per segment is computed via the following,

$$T_i = exp(j(2\pi f t_i^*)) \quad \forall i \in \{1, \ldots, K\} , \tag{32}$$

where $T_i$ is the complex Fourier factor of time delay, and $f$ is the frequency that matches each frequency bin in the spectral transform.

The combination of the two estimations now reads, $\bar{V}_{xcorr} = \bar{V}_d + T \odot \bar{V}_g$. The time domain signal is obtained as before, using the inverse short-time Fourier transform. $\bar{v}_{xcorr} = iSTFT(\bar{V}_{xcorr})$.

**Qualitative Results**     A visualization of $\bar{v}_d$ and $\bar{v}_g$ is depicted in Fig. 3. Evidently, both signals have a phase difference, which changes over time. For example, in panel (b), at [0.52,0.56] the phase of the generative estimation tracks the phase of the deterministic model. At the other end, at [0.56,0.64], the phase between the signal is offset by $\pi$. Recall that while $\bar{v}_d$ is given the full information about the phase, $\bar{v}_g$ is given only magnitude information and thus the absolute phase is not synced. Therefore, the alignment procedure is necessary.

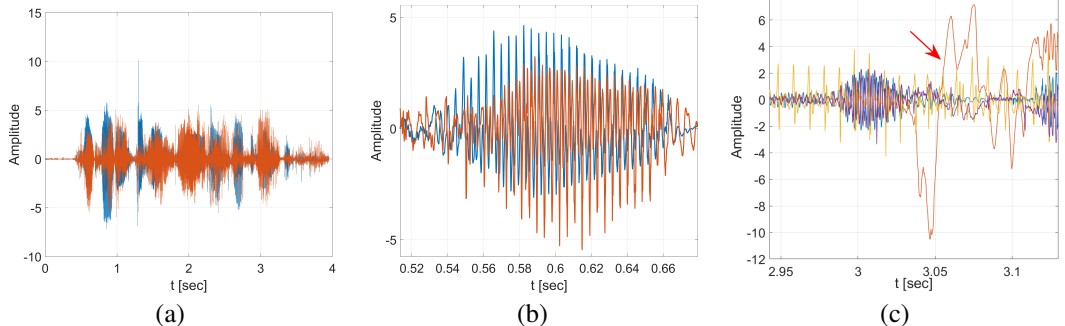

(a)         (b)         (c)

Figure 3: (a) Visualization of $\bar{v}_d$, and $\bar{v}_g$, (b) A zoom-in that shows the phase difference between two estimations. Blue-$\bar{v}_d$, Orange-$\bar{v}_g$. (c) An example where $\bar{v}_g$ is more precise than $\bar{v}_d$, and the resulting combination that tracks the source $v$. Blue-$v$, Orange-$\bar{v}_d$, Yellow-$\bar{v}_g$, Purple-$\bar{v}$.

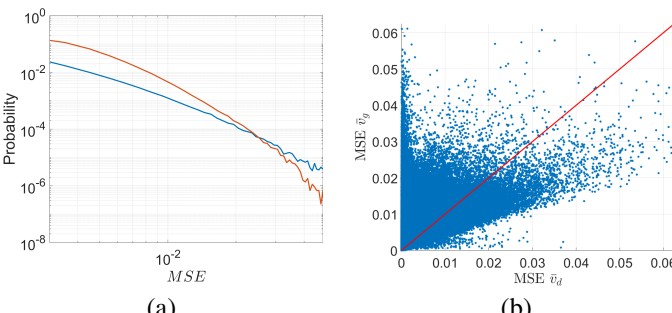

(a)         (b)

Figure 4: Comparing the error rates of the deterministic and the generative parts. (a) The histograms of the Mean Square Error (MSE) between $v$ to $\bar{v}_d$ (Blue), and $v$ to $\bar{v}_g$ (Orange). (b) Scatter plot of the MSE for $\bar{v}_d$ (x-axis) vs. $\bar{v}_g$ (y-axis).

The generated signal, $\bar{v}_g$, does not track the deterministic part, $\bar{v}_d$, one-to-one, but sometimes fixes sudden errors of other sources that the deterministic model could not handle correctly. This phenomenon is depicted in Fig. 3(c), where the deterministic signal has some bursts. As can be seen, the combined signal is more related to the ground truth signal.

The supplementary presents samples of both the deterministic and the generative estimation. While the deterministic estimation presents less background noise, it misses multiple temporal parts, which are filtered out by the consecutive chain of static filters of the deterministic model. The signal produced by the generative model, on the other hand, maintains a considerable amount of background noise but is able to sample parts that were previously omitted. The combination of both estimations yields a signal that is closer to the desired source.

To estimate the relative quality of the deterministic and generative estimations, the Mean Square Error between the different estimations and the aligned sources is calculated in short segments of 20[ms]. The histogram depicted in Fig. 4(a) shows that the generative signal $\bar{v}_g$ has a slight tendency to greater error, but has the same performance overall. This is in agreement with Eq. 16, where we expect that both $\bar{v}_g$, and $\bar{v}_d$ obtain similar performances. This result confirms that a combination of both estimations should improve the overall estimation of sources. A scatter plot of the errors is depicted in Fig. 4(b). The red line indicates a ratio of one to one between the deterministic error and the generative one. As can be seen, the error of the generative estimation is within the same range as that of the deterministic estimation, and there is no estimation that is markedly better than the other.

**Source separation results** The voice separation results for all the benchmarks are depicted in Tab. 1. For the WSJ0 dataset, when separating a mixture of two sources, the improvement in terms of SI-SDR is 1.5dB over the current state-of-the-art. Using our method not only improved the SI-SDR beyond all other methods but also broke the previous bound for deterministic models. This emphasizes the strength of the diffusion models as holding independent modeling benefits that can be used to improve speech separation. For three sources, separation is improved by 1dB, again obtaining state-of-the-art results, but somewhat lower than the classical bound of deterministic methods. We note that the ablation that removes network $F$ and introduces a heuristic alignment and equal mixing performs considerably better than the deterministic part (the SepFormer baseline) but only slightly better than the iterative SepIt. The deterministic HiFiGAN ablation shows little to no improvement,

Table 1: The performance obtained by our method on the various WSJ0 and LibriSpeech benchmarks. The reported values are SI-SDRi $[dB]$ (higher is better). -F refers to cross-correlation ablation, where the mixing network $F$ is not used. Mean of 5 is the mean signal of 5 runs with different noise initializations. *These missing runs of applying our method to baseline models are due to the inability to obtain the pre-trained baseline models.

| | WSJ0 | | LibriSpeech | | | |
|---|---|---|---|---|---|---|
| Method | 2Mix | 3Mix | 2Mix | 5Mix | 10Mix | 20Mix |
| Classical Upper Bound (Lutati et al., 2022) | 23.1 | 21.2 | 23.1 | 14.5 | 12.0 | 8.0 |
| Generative Upper Bound (ours) | 26.1 | 24.2 | 26.1 | 17.5 | 15.0 | 11.0 |
| DiffSep (Scheibler et al., 2022) | 14.3 | - | - | - | - | - |
| SepIt (Lutati et al., 2022) | 22.4 | 20.1 | - | 13.7 | 8.2 | - |
| SepFormer (Subakan et al., 2021) | 22.3 | 19.8 | 20.6 | - | - | - |
| SepFormer + HiFiGAN (Kong et al., 2020a) (ablation) | 22.3 | 20.0 | - | - | - | - |
| SepFormer + DiffWave -F (ablation) | 22.6 | 20.3 | 20.8 | - | - | - |
| SepFormer + DiffWave (ours) | 23.9 | 20.9 | 21.5 | - | - | - |
| SepFormer + mean of 5 DiffWave (ours) | **24.1** | **21.0** | **21.9** | - | - | - |
| SepFormer + UnivNet | 24.0 | 20.8 | 21.6 | - | - | - |
| Gated LSTM (Nachmani et al., 2020) | 20.1 | 16.9 | - | 12.7 | 7.7 | 4.3 |
| Gated LSTM + DiffWave -F (ablation) | -* | -* | - | 13.0 | 8.1 | 4.5 |
| Gated LSTM + DiffWave (ours) | -* | -* | - | 14.2 | 9.0 | 5.2 |
| Gated LSTM + mean of 5 DiffWave (ours) | -* | -* | - | **14.4** | **9.3** | **5.5** |

supporting the divide we make between deterministic and nondeterministic models. However, the non-deterministic GAN-based UnivNet Jang et al. (2021) provides results that are similar to those obtained with DiffWave, further strengthening our claim that non-deterministic methods can improve the performance of deterministic ones.

For LibriSpeech we use an available SepFormer model as a baseline in the case of two speakers, and for the other mixtures, an available Gated LSTM model. As can be seen, for all numbers of sources, a major improvement is obtained over the current state of the art. The alignment ablation degrades the separation result and is inferior to the learned method, but is still competitive. Evidently, there is still room for improving separation methods. For five speakers, the result obtained falls 0.3dB short of surpassing the classical upper bound. For other mixtures, the gap is larger.

Since the output of the diffusion process is conditioned on the random seed used, the option to run the randomized part multiple times is readily available. To explore this, we present the results obtained by simply averaging the output of five runs of the DiffWave with different noise initializations. As can be seen in the table, this leads to a small but consistent improvement, which is, as expected, still lower than the generative upper bound. Another question is the suitability of the method for working with noisy datasets, which is explored in Appendix C. As shown, our method can improve the state of the art method on that benchmark as well.

## 6 CONCLUSIONS

A general upper bound for source separation using generative models is proposed. The generalization of the upper bound suggests that for a pretrained generative model, a maximal improvement of 3dB from the previous deterministic bound is achievable. In addition, a simple yet effective method is suggested, where the deterministic signals and the generated signals are combined in the frequency domain. The combination procedure is learned and shown to be superior to the classical alignment method. When tested on various numbers of speakers, the estimation always yields improvement. For two sources, the separation is able to surpass the previous upper bound, demonstrating the need to go beyond a deterministic model.

The upper bound derived here is general in the sense that a similar bound would hold when the underlying distributions are replaced to fit other data domains. This is left for future work.

## 7 ACKNOWLEDGMENTS

This work was supported by a grant from the Tel Aviv University Center for AI and Data Science (TAD), and the Blavatnik Family Foundation. The contribution of Shahar Lutati is part of a Ph.D. thesis research conducted at Tel Aviv University.

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

Table 2: The performance obtained by our method on the WSJ0. The reported values are SI-SDRi $[dB]$.

| Method | WHAM! WSJ02Mix |
|---|---|
| SepFormer | 16.3 |
| SepFormer + DiffWave (ours) | **16.8** |

## A  PROOF OF LEMMA 3.2

The proof of Lemma 3.2 in the main text is below.

*Proof.* Using the definition of mutual information, and chain rule for entropy

$$
\begin{aligned}
I(X;Y,Z) &= H(Y;Z) - H(Y;Z|X) \\
&= H(Z) + H(Y|Z) - H(Z|X) - H(Y|X,Z) \\
&= (H(Z) - H(Z|X)) + (H(Y|Z) - H(Y|X,Z)) \\
&= I(X;Z) + I(X;Y|Z)
\end{aligned}
\tag{33}
$$

$\square$

## B  SAMPLE SPECTROGRAMS

Figure 5 presents sample spectrograms. Shown are the source, the deterministic estimation, the estimation obtained by the generative model, and their combination in the frequency domain using the coefficients produced by the function $F$.

## C  ADDITIONAL RESULTS

### C.1  RESULTS ON THE WHAM! BENCHMARK

The WHAM! dataset is a collection of two-speaker mixtures with background noise, designed to be a challenging benchmark for speech separation systems. The dataset was created by mixing speech from the WSJ0 dataset with noise from the WHAM! noise dataset, which contains recordings of urban environments such as restaurants, cafes, bars, and parks. In Tab. 2 utilizing a pretrained SepFormer model and a pretrained DiffWave, our method is able to surpass current state of the art result.

### C.2  ADDITIONAL ABLATIONS

To assert that the additional parameters in the alignment network are not the lead cause for the improvement, five ablations of running the alignment network over the generative output alone or the deterministic output alone are done: (a) without aligning the phase to the source signal, (b) Linearly aligning of the phase, (c) using the alignment network with $v_d$ twice, (d) using the alignment network with $v_d$ and phase-shifted $v_d$, and (e) retraining the alignment network with only the absolute phase without the difference of the phase and then running as in our original method.

The reported results are in Tab. 3. In all single module variances (a-d), the performance of the reconstruction is worse than the reconstruction using the combination of both the deterministic and generative samples. Ablation (e) demonstrates that the phrase difference is an important input to the alignment network.

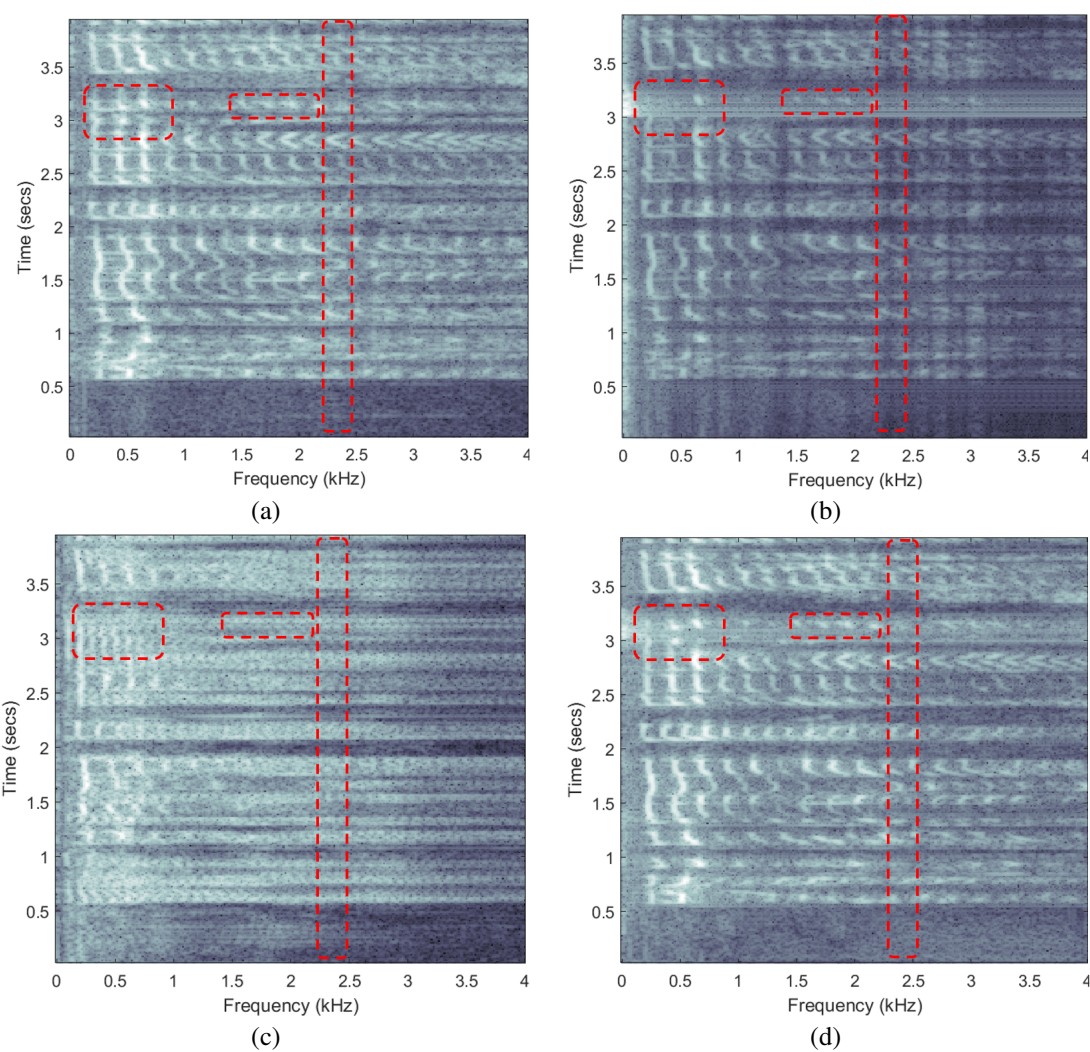

Figure 5: (a) Spectrogram of a source. (b) Spectrogram of the deterministic estimation $\bar{\boldsymbol{v}}_d$ (c) Spectrogram of the generated voice $\bar{\boldsymbol{v}}_g$ (d) Spectrogram of the combined estimation $\bar{v}$.

Table 3: Ablations over Libri2Mix and Libri5Mix. The reported values are SI-SDRi $[dB]$.

| Method | Libri2Mix | Libri5Mix |
|---|---|---|
| $\bar{\boldsymbol{v}}_d$ (baseline) | 20.6 | 12.7 |
| (a) $\bar{\boldsymbol{v}}_g$ | 19.8 | 10.2 |
| (b) $\bar{\boldsymbol{v}}_g$ phase aligned | 20.0 | 11.3 |
| (c) only $\boldsymbol{v}_d$ w/ alignment network | 20.2 | 12.4 |
| (d) only $\boldsymbol{v}_d$ shifted | 20.3 | 12.6 |
| (e) re-trained without phase difference | 21.3 | 13.7 |
| Our full method | 21.5 | 14.2 |

