# OpenReview forum: "Separate and Diffuse: Using a Pretrained Diffusion Model for Better Source Separation"
_ICLR.cc/2024/Conference — ICLR 2024 poster_

### Official Review · Reviewer_9rFE · 2023-10-18

**Soundness:** 3 good
**Presentation:** 3 good
**Contribution:** 2 fair
**Rating:** 6
**Confidence:** 5

**Summary:**

This paper presents a single-channel speech separation method using a combination of deterministic and generative models. An upper bound on the signal distortion ratio (SDR) of this combined method is derived, suggesting that it has the potential to be better than the deterministic model alone.

**Strengths:**

1. The authors innovatively integrate deterministic models with generative models for speech separation and theoretically demonstrate the performance boundaries of this approach.
2. To address the phase shift issue between the output estimates of the deterministic and generative models, an alignment network is employed to estimate two parameters for the fusion of the outputs from both models.
3. Results across multiple datasets and models indicate that this method can further improve the performance of existing models.

**Weaknesses:**

1. **Generative Model Selection:** The authors' choice of utilizing only one diffusion-based generative model to validate the performance enhancement brought by noise introduction appears to be limiting. Although the theoretical incapacity of deterministic generative models to achieve performance enhancement has been demonstrated, the naturalness in generation by HiFiGAN is inherently lower than that of DiffWave. This raises concerns about whether this disparity is the reason for HiFiGAN's lack of improvement. I would recommend the authors consider incorporating other diffusion-based generative models, such as FastDiff [1], or superior deterministic models like UnivNet [2], to bolster the robustness of their results.
2. **Deterministic Model Upper Bound:** I disagree with the notion that deterministic models possess an upper bound. Recently, TF-GridNet results surpassed the so-called upper bound of deterministic models, attaining an SI-SDRi of 23.4, which is strikingly close to the SepFormer + DiffWave model. It would be prudent for the authors to include results on TF-GridNet to underscore the necessity of noise introduction in diffusion-based generative models. Training a generative model alone can be computationally demanding, and might not be the optimal solution just for a marginal performance gain.
3. **Alignment Network Concerns:** Regarding the alignment network, the authors utilize the relative phase difference between $V_g$ and $V_d$ as well as the phase of $V_d$ as inputs to align the phase of the fused output with $V_d$. This poses a question: Is the observed enhancement in model performance a result of the phase alignment between $V_g$ and $V_d$, or is it due to the introduction of additional parameters, i.e., the alignment network? Another hypothesis worth considering is if the alignment network, when directly using the phase of $V_g$ and $V_d$ as inputs, would produce a similar effect.
4. **Testing on Noisy Datasets:** One notable observation from the manuscript is its primary focus on clean datasets for evaluation. It would be beneficial to see how the proposed combined model performs on noisy datasets, such as WHAM! or the noisy version of Librimix. Evaluating on these datasets can provide insights into the model's robustness in more realistic scenarios, where environmental noise might significantly impact the performance of the generative model. Such an evaluation will offer a more comprehensive understanding of the model's real-world applicability and its ability to tackle inherent challenges posed by noisy environments.
5. **Training Concerns:** The manuscript should clearly specify whether the separation and generative models were involved in the training of the alignment network $F$.
6. **Symbol Representation:** Please ensure a consistent and standardized representation of symbols throughout the paper. Conventionally, vectors are denoted in boldface.

[1] Huang R, Lam M W Y, Wang J, et al. Fastdiff: A fast conditional diffusion model for high-quality speech synthesis[J]. arXiv preprint arXiv:2204.09934, 2022.

[2] Jang W, Lim D, Yoon J, et al. Univnet: A neural vocoder with multi-resolution spectrogram discriminators for high-fidelity waveform generation[J]. arXiv preprint arXiv:2106.07889, 2021.

**Questions:**

My detailed questions are as described above.

---

> ### Author Response · Authors · 2023-11-10
> **Thank you for your very insightful feedback, which we will address as soon as possible + a quick note**
>
> We would like to promptly address the second item, which suggests our work might lack theoretical soundness
>
> As the reviewer correctly noted, TF GridNet achieved a 23.4dB gain with SISDR, while remaining deterministic. However, as Lutati et al. (2022) highlighted, this upper bound is valid only when each chunk is processed independently of the others. They write:
> >“A clear limitation of the bound is that it holds only for a network D that jointly processes i.i.d stationary segments. This is not the case if a neural network processes the entire signal without segmentation.“
>
> It is important to note that TF GridNet's gain is influenced by the signal's length due to time-frequency ambiguity, while in real-world applications, signals often vary in length.
>
> We will highlight this limitation in the revised version of our paper.

---

> ### Author Response · Authors · 2023-11-21
>
> We thank the reviewer for the comprehensive feedback.
>
> *Generative model selection:*
>
> Our proposed method relies on conditioning of the generative signal based on deterministic estimation of the signal.
>
> FastDiff’s conditioning is based on text and thus is not in the scope of this work.
>
> UnivNet is a non-deterministic GAN, which has a similar input to that of the diffusion-based model that we use (DiffWave). It is indeed very interesting to evaluate since the GAN we employed in the paper (HiFiGAN) is deterministic and was employed to show that deterministic de-noising methods do not improve performance.
>
> Following the review, we have run UnivNet in combination with Gated LSTM and our phase alignment network. The results for Libri2Mix are reported in the table below (and added to Table 1 of the manuscript). Evidently, UnivNet and DiffWaveresults are within negligible differences of around 0.1dB. This strongly reinforces our main claim.
>
> *Deterministic Model Upper Bound:*
>
> As the reviewer correctly stated, our upper bound depends on that of Lutati et. al., which assumes sequential processing of uncorrelated chunks. The bound does not hold for the time-frequency approach of TF-GridNet. While TF-GridNet obtains SOTA results on WSJ2 for two speakers (only), it does not mean that chunking-based methods are inferior: (1) TF-GridNet is unable to deal with non-stationary segments, such as silence, which do not appear in WSJ2 but are frequent in the actual applications, (2) TF-GridNet needs to be retrained for every signal length, and even worse (3) TF-GridNet can only handle relatively short signals. It is also worth noting that (4) time-frequency methods have a sharp decrease in performance as the sample frequency drops since the frequency domain estimation (using FFT) becomes blurred. Finally, (5) TF-GridNet was not shown to be effective for more than two speakers.
>
>
> *Alignment Network Concerns*
>
> Following the review, we have conducted three additional ablations that are added to the revised transcript:
> Using the alignment network with $v_d$ twice.
> Using  the alignment network with $v_d$ and phase-shifted $v_d$
> Retraining the alignment network with only the absolute phase without the difference of the phase. Then running as in our original method.
>
> In all tests, the results of the ablation were worse,
> | Ablation Type                       | Libri2Mix | WSJ2MIX | Libri5Mix |
> |-------------------------------------|-----------|---------|-----------|
> | (a) Only $v_d$                            | 20.2      | 22.0    | 12.4      |
> | (b) Only $v_d$ shifted                    | 20.3      | 22.1    | 12.6      |
> | (c) Re-trained without phase difference | 21.3      | 23.3    | 13.7      |
> | Our Method                          | 21.5      | 23.9    | 14.2      |
>
> As can be seen, the phase alignment network does not improve performance when the generative model’s output is left out.
>
> The alternative network architecture in (c) is somewhat inferior. In this case, the phase difference information exists only implicitly (i.e., it can be inferred from the two absolute phases but is not given explicitly).
>
> *Testing on Noisy Datasets:*
>
> Following the review, we evaluated our method on the WHAM! benchmark. We used a pretrained version of SepFormer over WHAM found in hugging faces.
>
> | Method                      | WSJ2Mix + WHAM |
> |-----------------------------|----------------|
> | Sepformer                   | 16.3           |
> | SepFormer + DiffWave (ours) | 16.8           |
>
> As a side note: the additional independent noise in WHAM can be modeled, and a tighter upper bound can be computed for this dataset, using the insights of Fig. 1.
>
> *Training Concerns:*
>
> Both pretrained networks are not fine-tuned and are not part of the training of the alignment network.
>
> *Symbol Representation:*
>
> We thank the reviewer and we will make sure that the symbols are consistent.

---

> > ### Comment · Reviewer_9rFE · 2023-11-21
> > **Response authors**
> >
> > For your response regarding TF-GridNet, I have some differing views.
> >
> > 1. I don't believe that TF-GridNet cannot handle audio of varying lengths because, to my knowledge, the lengths of the data in the WSJ0 test set are also different. If you have conclusive evidence, please provide relevant literature or experimental results.
> >
> > 2. Regarding the FFT method in the time-frequency domain, we can achieve better performance by setting appropriate window lengths and strides. Moreover, in reverberant situations, time-frequency domain methods often perform better than time-domain methods.
> >
> > 3. As for the issue of not handling segments with silence, this requires evidence to be claimed. When datasets include similar scenarios, I believe the model can also learn similar separation paradigms.
> >
> > I also have a concern about the complexity and the number of parameters of the method proposed in the paper. Compared to deterministic models, these kinds of cascaded models often require more parameters and complexity, and I wonder whether this is necessary for enhancing performance.
> >
> > Lastly, I thank the authors for providing additional experiments and analyses to address my concerns. I have decided to raise my score to 6.

---

### Official Review · Reviewer_d285 · 2023-10-31

**Soundness:** 4 excellent
**Presentation:** 3 good
**Contribution:** 2 fair
**Rating:** 6
**Confidence:** 4

**Summary:**

This paper presents a new method for the source separation problem. First, a discriminative model is used to produce output sources. Then a generative model is used to refine those signals conditioned on the output of the discriminative model. Finally, a learned mixing coefficient is used to blend between the generative and discriminative outputs.

**Strengths:**

Overall I appreciate the method and the author's approach to using generative models. Generative models have shown strong performance in many areas and their use in source separation has been somewhat limited. I also appreciate the theoretical analysis which provides solid justification for the choices and results.

The ablation study shows that the the mixing network is in fact helpful, since the naive approach would be to just use the generative output v_g directly.

It is also nice that the authors use a variety of discriminative models and compare them, which shows that the method is general.

The output audio examples provide a good sense to the listener of the model's performance

**Weaknesses:**

The main issue I have is the usefulness of the theoretical bounds given the underlying assumptions. The paper build heavily on the analysis in Lutati et al. where the bounds were derived by making assumptions on the context used. These assumptions provide a bound that is not realistic, as evidenced by the fact that the bound for WSJ2 mix is 23.1dB but TF-Gridnet achieved 23.4dB gain using purely a discriminative complex valued model. This is something that should be discussed in the paper more.

**Questions:**

I would like to see an ablation where only the generative output is used after conditioning on the discriminative output (not a simple average like the current ablation). Have you done those experiments and how did they perform?

---

> ### Author Response · Authors · 2023-11-21
>
> We thank the reviewer for the detailed feedback.
>
> As the reviewer correctly stated, our upper bound depends on that of Lutati et. al., which assumes sequential processing of uncorrelated chunks. The bound does not hold for the time-frequency approach of TF-GridNet. While TF-GridNet obtains SOTA results on WSJ2 for two speakers (only), it does not mean that chunking-based methods are inferior: (1) TF-GridNet is unable to deal with non-stationary segments, such as silence, which do not appear in WSJ2 but are frequent in the actual applications, (2) TF-GridNet needs to be retrained for every signal length, and even worse (3) TF-GridNet can only handle relatively short signals. It is also worth noting that (4) time-frequency methods have a sharp decrease in performance as the sample frequency drops since the frequency domain estimation (using FFT) becomes blurred. Finally, (5) TF-GridNet was not shown to be effective for more than two speakers.
>
> > I would like to see an ablation where only the generative output is used after conditioning on the discriminative output (not a simple average like the current ablation). Have you done those experiments and how did they perform?
>
> In the paper, we demonstrate the need to average the output of the generative model with the deterministic method’s output in Fig. 3, 4.
>
> Following the review, we have added to the appendix the results for Libr2iMix and Libri5Mix obtained without mixing the generative model output. These are provided twice:
>
> Without aligning the phase to the source signal.
> Linearly aligning of the phase.
>
> In both cases, the performance of the reconstruction with the generative part alone (of course, conditioned on the deterministic part) is worse than the reconstruction using the combination of both the deterministic and generative samples.
>
> | Method            | Libri2Mix | Libri5Mix |
> |-------------------|-----------|-----------|
> |$\bar v_d$ (baseline)|             |              |
> | (a) $\bar v_g$          | 19.8      | 10.2      |
> | (b) $\bar v_g$ phase aligned | 20.0      | 11.3      |
> | Our full method        | 21.9      | 14.2      |
>
>
> Evidently, the generative method’s output is not as effective as the deterministic one (reinforcing figures 3 and 4) but improves the performance when combined with it. We note that the theoretical analysis assumes this combination. $\bar v_g$ by itself has the same bound as $\bar v_d$. Only their sum is improved.

---

### Official Review · Reviewer_fbeS · 2023-11-05

**Soundness:** 3 good
**Presentation:** 3 good
**Contribution:** 3 good
**Rating:** 6
**Confidence:** 2

**Summary:**

This paper proposes a diffusion-based post-processing module for single-channel speech enhancement. The authors present a mathematical derivation of the upper-bound of the source-to-distortion for generative methods, proving an improvement over the bound derived for deterministic models in prior work. They also present an architecture that combines the discriminative estimation and the generative estimation in the Fourier domain, which consists of a separation module, a generative module, and a mixing weights prediction module. Empirical results on a number of popular speech separation architectures on two speech separation datasets with multiple speaker numbers demonstrate the effectiveness of the proposed approach.

**Strengths:**

The idea of diffusion-based separation has been applied in speech separation, but the primary novelty of this work lies in the mathematical perspective of improving the SDR upper-bound with the generative approach by combining the output of the discriminative and generative estimations.

The experiments are conducted across several different separation architectures and for two popular source separation datasets (with several speaker-number settings), and an ablation study of the mixing network is performed. The empirical results demonstrate the improvement of the proposed method with deterministic SOTA on speech separation.

**Weaknesses:**

My major concern about the current version of this manuscript is the clarity of the writing. There are a number of notations (e.g., $v_r, v_{gr}, v_{dr}$) that are used across multiple sections of the paper, but these notations are not easy to follow and the consistency could be improved. In particular:

- Introduction could be clearer with all variables properly defined with types (real vs complex), and dimensions. Additionally, I suggest beginning with some motivation (reiterating parts of Section 2) but focusing on the bottleneck of the existing (deterministic and discriminative) approaches.

- Please make sure to define the acronyms at the first usage (e.g., SDE in Section 2, GM in Section 3.1).

- Please resolve the inconsistency
  - notations between the opening paragraph ([$\alpha_i, \beta_i$] = F($\bar{v}_d^i, \bar{v}_g^i$) vs ([$\alpha_i, \beta_i$] = F($\bar{V}_d^i, \bar{V}_g^i$) in (4).
  - $I(m_r, v_r)$ --> $I(m_r; v_r)$ in (6).
  - $p(v_{d}r) --> p(v_{dr})$ in Section 3.1.
  - The notations of $v_{gr}, \bar{v}_{gr}$ and $v_{dr}, \bar{v}_{dr}$ in Section 3.

- In (10): it seems there is overloading of the notation $p(v_{gr})$ on LHS and RHS of the equation.

- The font size of the equations and figure labels could be improved.
  - In Figure 4 (a), the x-label "MSE" is in italics, whereas for (b) it is not.
  - I would recommend disabling the italics for function names such as "argmax", "ELBO", "SDR", "log", etc.

**Questions:**

- I'm uncertain on how the two inequalities in (11) are derived. For the first inequality, how is $I(v_r; v_{gr})$ related to $I(v_r; v_{dr},v_{gr})$ in (9)? The necessary condition for the second inequality is $I(v_r; v_{gr}, v_{dr}) \leq I(v_r; m_r)$, but this is not implied from (7) or (9). It would be helpful if the authors could clarify the steps. Also, it'll be helpful to explain how the "3.0" db is obtained in (21).

- Any reason for transposing the horizontal and vertical axes for the visualizations in Figure 5? It is conventional to display the spectral information in the vertical axis and temporal dimension horizontally.

Update after rebuttal: I'd like to thank the authors for addressing the questions. The scores have been updated.

---

> ### Author Response · Authors · 2023-11-21
>
> We thank the reviewer for the comprehensive feedback.
>
> The manuscript has been revised following the requested changes.
>
> $GM$ is defined in the introduction and then again in the beginning of the analysis section. SDR  in definition 3.1.
>
> To answer the reviewer’s questions:
>
> 1. $I(v_r;v_{gr})$ is the mutual information between the source at the r-th chunk and the generative sample in the same chunk. $I(v_r;v_{dr},v_{gr})$ is the mutual information between the source and both the deterministic estimation and the generative sample. The first term is upper bounded by the second part by definition since additional signals can add information and not remove information.
>
> 2. In Eq. 9 we show that the mutual information between the generative and source is upper bound by the sum of the mutual information of the source and mixture (first term) and the mutual information between the (1) source, and (2) the tuple of (generative sample, deterministic estimation). Since the generative sample is from approximately the same distribution as the deterministic distribution, the same data-processing theorem bound holds for it. Using Eq. 7 we can upper bound again the second term in Eq. 9  to obtain the second inequality.
> The 3dB is due to the factor of 2 over the previous bound which translates to a 3dB addition in the logarithmic scale.
>
> In the final version, we will recreate Fig. 5 of the appendix, replacing the two axes as requested.

---

### Meta-Review · Area_Chair_wZNy · 2023-12-04

**Metareview:**

This paper introduces a diffusion-based post-processing module for enhancing single-channel speech. The authors derive an upper-bound for the source-to-distortion ratio for generative methods, showing an improvement over the bound derived for deterministic models in previous work. They propose an architecture that merges discriminative and generative estimation in the Fourier domain. The effectiveness of this approach is demonstrated through empirical results on several popular speech separation architectures across two speech separation datasets with varying speaker numbers.

Three reviewers consistently recommended marginal acceptance of the paper. The authors addressed the concerns raised by reviewers.

**Justification For Why Not Higher Score:**

The paper’s contributions do not appear to be significant enough, and the methods proposed lack sufficient novelty for a higher score.

**Justification For Why Not Lower Score:**

It is commendable to see the mathematical perspective applied to improve the upper-bound of SDR using a generative approach.

---

### Decision · Program_Chairs · 2024-01-16

Accept (poster)